# Mapping Protein Targets of Carnosol, a Molecule Identified in *Rosmarinus officinalis*: In Silico Docking Studies and Network Pharmacology

**María Taboada-Alquerque** [1], **Danilo Pajaro-Valenzuela** [1], **Karina Caballero-Gallardo** [1], **Alejandro Cifuentes** [2], **Elena Ibáñez** [2], **Maicol Ahumedo-Monterrosa** [3], **Elena E. Stashenko** [4] **and Jesus Olivero-Verbel** [1,*]

[1] Environmental and Computational Chemistry Group, School of Pharmaceutical Sciences, Zaragocilla Campus, University of Cartagena, Cartagena 130015, Colombia
[2] Laboratory of Foodomics, Institute of Food Science Research, CIAL, CSIC, Nicolás Cabrera 9, 28049 Madrid, Spain
[3] Natural Products Group, School of Pharmaceutical Sciences, Zaragocilla Campus, University of Cartagena, Cartagena 130015, Colombia
[4] Center for Chromatography and Mass Spectrometry, CROM-MASS, CIBIMOL-CENIVAM, Industrial University of Santander, Carrera 27, Calle 9, Building 45, Bucaramanga 680002, Colombia
[*] Correspondence: joliverov@unicartagena.edu.co; Tel.: +57-(5)-6698179 or +57-(5)-6698180; Fax: +57-(5)-6698323

**Abstract:** Carnosol is a natural diterpene present in *Rosmarinus officinalis* L. (rosemary) with anti-tumor and anti-inflammatory properties. Despite its importance, the pharmacological mechanisms underlying the interactions between carnosol and human targets are still unclear. The goal was to identify plausible human target for carnosol and the network pharmacology. Rosemary was analyzed using HPLC-QTOF-MS/MS. Potential carnosol targets were identified using docking and a public database (CTD). Carnosol was screened against 708 human proteins using AutoDock Vina, and affinity values were used as prioritization criteria. The targets set was uploaded to WebGestalt to obtain Gene Ontology (GO) and KEGG pathway enrichment analysis. HPLC-QTOF-MS/MS analyses allowed the tentative annotation of nine chemicals, with carnosol being the most ionized. There were 53 plausible targets for carnosol, with 20 identified using virtual screening, including Hsp90$\alpha$ ($-10.9$ kcal/mol), AKR1C3 ($-10.4$ kcal/mol), and Hsp90$\beta$ ($-10.4$ kcal/mol), and 33 identified from CTD. The potential targets for carnosol identified with PPI and molecular docking were HSP90AA1, MAPK1, MAPK3, CAT, JUN, AHR, and CASP3. GO terms and KEGG pathways analysis found that carnosol is closely related to infection (Chagas, influenza A, toxoplasmosis, and pertussis) and inflammation (IL-17 and TNF signaling pathway and Th-17 cell differentiation). These results demonstrated that carnosol may induce an immuno-inflammatory response.

**Keywords:** *Rosmarinus officinalis*; carnosol; docking; network pharmacology; inflammation

## 1. Introduction

*Rosmarinus officinalis* L. (rosemary) is a plant commonly employed in food and beverage industries [1], as well as in personal care, nutrition, and health sectors [2]. Rosemary is cultivated in many parts of the world, mainly in Mediterranean and South American countries, becoming one of the species of the Lamiaceae family with the greatest economic impact [3]. The metabolome of these plants has been well characterized and includes polyphenols [4] and terpenes [5], such as carnosol, carnosic acid, rosmanol, and rosmarinic acid [6]. Extracts or individual compounds isolated from this plant have shown antioxidant [7], anti-inflammatory [8], antidiabetic [9], anti-obesity [10], and antitumor effects [11]. The antioxidant capacity of this plant has been mostly attributed to diterpenes, including carnosol, once of compounds that represent more than 90% of the antioxidant pool of their extracts [12].

Carnosol is an *ortho*-diphenolic diterpene obtained as an oxidation product from carnosic acid, a natural benzenediol abietane terpene [13] with several biological advantages, such as being able to alleviate inflammation and control cell proliferation, with a marked inhibitory effect on the growth of various types of cancer cells, such as B16F10 melanoma [14] and human colon adenocarcinoma cells [15]. The chemical structure of carnosol is shown in Figure 1.

**Figure 1.** Structure of carnosol.

The antitumor activity of this molecule has been attributed to its ability to induce apoptosis, inhibit cell cycle division, and other molecular mechanisms that modulate biochemical processes associated with proliferation [16]. However, the primary molecular targets used by this chemical are not well known. In recent years, the association between molecular docking studies and pharmacological networks has been a synergistic approach effectively applied for the prediction of target proteins and biochemical pathways activated by drugs [17]. This has allowed the exploration of protein–ligand interactions, as well as their associated signal transduction pathways through bioinformatic analysis, associating a number of proteins targeted by a ligand through molecular docking, with data from in vitro, in silico, and in vivo studies, creating biological maps for molecular functions, cellular components, and pathways. As a result, the approach has been successfully applied in the medical context to identify the mechanisms of action of natural compounds used for the treatment of diverse pathologies [18].

This study aimed to identify plausible human targets for carnosol using in silico protocols as well as network pharmacology tools to uncover underlying molecular mechanisms mediated by carnosol targets. Thus, compound–target–pathway networks were constructed to investigate the involved mechanisms of action from a macroscopic perspective.

## 2. Materials and Methods

A virtual screening was carried out to explore the capacity of carnosol, the main chemical compound identified in the hydro-alcoholic extract of rosemary traded in the local market in Cartagena, Colombia, to target human genes involve in different biological functions and pathways.

### 2.1. Analysis of the Hydro-Alcoholic Extract of Rosemary by HPLC-QTOF-MS/MS
2.1.1. Plant Material and Extraction

The leaves of *R. officinallis* were purchased at the local market in Cartagena, Colombia. The leaves extract was obtained according to a previously reported methodology [19]. Leaves of *R. officinallis* were freeze-dried, crushed, and the powder (50 g) was mixed (1:10, *w/v*) with 70% ethanol in water for 24 h at room temperature under darkness. The extract was filtered, and the solvent evaporated in a rotary evaporator at less than 45 °C at 100 rpm. The extract was freeze-dried until the dried sample was obtained. The product was stored at −20 °C until analysis.

2.1.2. Sample Preparation and HPLC-QTOF-MS/MS Conditions

The dried rosemary extract was diluted in an acetonitrile:water mixture (1:1) to obtain a concentration of 1 mg/mL. Samples were shaken for 5 min in a vortex and then centrifuged at 6000× *g* (8 min). The supernatants were filtered with filters of 0.20 μm particle size and transferred to autosampler vials. High-performance liquid chromatography/quadrupole time-of-flight mass spectrometry (HPLC-QTOF-MS/MS) was utilized to separate and characterize the components. Chromatographic separation was performed using a 1260 Infinity HPLC (Agilent Technologies, Santa Clara, CA, USA) equipped with a Variable Wavelength Detector (VWD) employing a InfinityLab Poroshell 120 EC-C18 column of 4.6 × 100 mm, ×2.7 μm particle size (Agilent Technologies, Santa Clara, CA, USA) with a mobile phase flow rate of 0.3 mL/min, consisting of water (A) and acetonitrile (B), with 0.1% HCOOH. Analysis started with 95:5 A:B, held for 1 min, followed by changed linearly up to 5:95 in 9 min, with a hold to 4 min; Then, changed to 100% acetonitrile in 1 min, and it was stable for 3 min. Column re-equilibration was performed by returning to 95:5 A:B at minute 23 and holding until 26 min. The mass analysis was obtained using a Quadrupole-Time of Flight tandem mass spectrometer 6530 Q-TOF detector (Agilent Technologies, Santa Clara, CA, USA), with Electrospray Ionization (ESI) operated in negative ion mode. The conditions for the mass detector were as follows: capillary voltage, +3.5 kV; nitrogen gas temperature, 320 °C; drying gas flow rate, 8.0 L/min; nebulizer gas pressure, 35 psig; fragmentor voltage, 135 V; skimmer, 65 V; and OCT RF, 750 V. MS/MS Data Acquisition mode was used to assist compound identification. Mass range in MS and MS/MS experiments were set at $m/z$ 100–1200 and 50–1200 at 3 spectra/s, respectively. MS and MS/MS data were collected using Agilent MassHunter Acquisition software (version 10.1). The data obtained were processed with the Agilent MassHunter Qualitative Analysis 10.0. Peak annotations were performed using the METLIN (metlin.scripps.edu, accessed on 15 January 2021) metabolite databases with a mass error of less than 5 ppm and manual dereplication. Compound identification was based on the exact masses and MS/MS spectra comparisons of the target compounds [19].

*2.2. Mapping Targets of Carnosol by Virtual Screening*

2.2.1. Preparation of Crystallographic Structures of Human Proteins for Molecular Docking

A group of 708 crystallographic structures of human proteins were downloaded from Protein Data Bank (PDB) (www.rcsb.org/pdb/home/, accessed on 10 June 2019) in pdb-formatted files, prepared, and optimized employing SYBYL-X 2.0 program package (Tripos, St. Louis, MO, USA). Water molecules and substructures were taken out, and side chain amides fixed. The protein optimization process for minimizing variables was carried out employing the Powell conjugate gradient algorithm, with the combined force fields Kollman united/Kollman All Atom, AMBER charges, a gradient convergence criterion of 0.005 kcal/mol, and a maximum of 1000 iterations. The structure optimized was saved in PDB format. The MGLTools 1.5.0 software was used to add polar hydrogens and convert PDB files to PDBQT format [20,21]

2.2.2. Preparation of Carnosol Structures

The 3D structure of carnosol (PubChem CID: 442009) [22] was downloaded from PubChem database https://pubchem.ncbi.nlm.nih.gov/ (accessed on 5 May 2021) [23] in sdf format, and optimized in Gaussian version 09 (Gaussian, Inc., Wallingford, CT, USA, 2009) using DFT/B3LYP methods and a set of 6–31G bases. The optimized geometries were then converted to mol2 and pdbqt with Open Babel version 2.3 [24].

2.2.3. Docking Calculations on Human Proteins

Docking molecular calculations were carried out using AutoDock Vina [25], configuring a box covering the entire protein and employing an exhaustiveness of 15 for all calculations [26] and 32 for top 20 carnosol targets [27]. The boxes were built applying a grid spacing of 1.0 Å. Calculations were performed by triplicate for each protein–ligand

complex, and the average presented as the affinity value (kcal/mol). The structure of the complex with the highest absolute affinity value was considered the best pose.

### 2.2.4. Molecular Docking Validation

The validation of the docking protocol was carried out by using 13 human protein structures targeted by carnosol. The approach consisted of removing the ligand from each X-ray structure and then re-docking it onto the protein using SYBYL-X 2.0 Package and AutoDock Vina. The experimental pose of the co-crystallized ligand on the PDB structure (www.rcsb.org/pdb/home/, accessed on 20 September 2021) was compared with that obtained after the re-docking procedure. The mean quadratic deviation (RMSD in Å) was employed as a criterion to validate the molecular docking protocol. Mean quadratic deviation values of less than 2 Å were considered good results [28].

### 2.3. Mapping Targets of Carnosol by Comparative Toxicogenomics Database (CTD)

Potential targets of carnosol were mapped using CTD (http://ctdbase.org/, accessed on August 2022), a database that provides information about genes, disease, phenotypes, and pathways related to small molecules. All this information was retrieved from the scientific literature until 2019 [29]. The number of interactions reported for each gene was used as a selection criterion. Genes with interaction $\geq 1$ were selected as candidate target genes.

### Docking Calculations on CTD Targets

Binding affinities for carnosol targets identified with CTD were calculated using the same protocol described previously, but with exhaustiveness of 32.

### 2.4. Docking Visualization

The residues on the binding site of proteins interacting with carnosol were identified using Protein–Ligand Interaction Profiler (PLIP) https://plip-tool.biotec.tu-dresden.de/plip-web/plip/index (accessed on 2 February 2023) [30]. For that purpose, the complexes were converted to pdb format to obtain pharmacophores that indicated the type and quantity of interactions formed in the protein–ligand interphase. The 3D structures of these complexes were visualized using the PyMOL version 1.4 [31].

### 2.5. Network Pharmacology

### 2.5.1. Genetic Ontology and Functional Interaction Pathway Enrichment

Genetic ontology (GO) analysis and KEGG pathway enrichment were carried out using the classification system of Gene Ontology and Kyoto Encyclopedia of Genes and Genomes (KEGG), providing an overview of the molecular characteristics, and enriched biological pathways associated with the carnosol target set. The enrichment analysis of GO terms for 20 proteins with better affinities for carnosol and for 33 targets from CTD were developed using web-based gene set analysis toolkit (WebGestalt). The significant results from the analysis were filtered using a *p*-value $\leq 0.05$. The pathways obtained with KEGG used by WebGestalt tool were visualized in the bar chart employing Matplotlib library in Python [32].

### 2.5.2. Pharmacological Network Analysis

The analysis of a pharmacological network helps to interpret the biochemical interactions among targeted proteins, likely suggesting the signaling transduction pathways intervened by the ligands. The protein–protein interaction (PPI) network for carnosol targets was built employing those 20 targets with the highest affinity values identified by virtual screening and the 33 targets identified in CTD. The PPI was built by uploading of targets set to Search Tool for the Retrieval of Interacting Genes (STRING, https://string-db.org/, accessed on 2 February 2023) database with the following set of parameters—specie: Homo sapiens, the minimum interaction score: 0.4. The PPI and carnosol—target—pathway

network were visualized using Cytoscape 3.6.1, and network properties were calculated by the Network Analyzer plug-in [33].

### 2.6. Molecular Dynamics Simulation

Molecular dynamics (MD) simulations were employed to explore the stability of carnosol in complex with respect to the potential targets selected by molecular docking. A MD simulation of 100 ns was performed using Gromacs version 2020.2 [34]. The forcefields used for the protein and the ligand were the CHARMM force field [35] and the CHARMM General Force Field (CGenFF) [36], respectively. Calculations included the protein model with and without the co-crystallized ligand and the complex potential target—carnosol. Complexes were immersed in a cubic periodic box in which each complex was solvated with a TIP3P water under periodic boundary conditions [37]. The systems were neutralized, and the ionic strength (0.1 mol L$^{-1}$) of the medium was adjusted by adding Na$^+$ and Cl$^-$ ions, keeping the number of particles constant. The energy minimization of the systems was performed until the energy converged. Next, an equilibrium phase was carried out keeping the pressure and temperature (NVT and NPT ensemble) constant at 300 K and 1.0 bar, respectively. The equilibration periods were 1.0 ns. The production runs were of 100 ns, and the trajectories were saved every 0.01 ns. Molecular dynamics results were used to calculate the root mean square deviation (RSMD), and the root mean square fluctuation (RMSF). MMPBSA.py script [38] in the AMBER 21 suite was used to predict the free binding energies of these protein–ligand complexes. In order to apply the MMPBSA.py script, the topologic file, coordinates file, and production file generated in Gromacs were converted to their counterparts in Amber. The interaction energy and solvation free energy for the complex, receptor, and ligand were used to obtain an estimate of the binding free energy according to MM/GBSA approaches [39].

### 3. Results

### 3.1. Analysis of the Hydro-Alcoholic Extract of Rosemary by HPLC-QTOF-MS/MS

The results of the HPLC-QTOF-MS/MS analysis for the hydro-alcoholic extract of rosemary are shown in Figure 2 and Table 1. The monoisotopic mass characteristics for negative ions of the compounds were tentatively identified, and fragment spectrum results (MS/MS) are presented in Supplementary Material (Table S1). The tentative annotations belong to various compounds classes, including phenolic acids (rosmarinic acid), flavones (isorhamnetin-rutinoside and dihydroxy-dimethoxyflavone), and diterpenes (rosmanol, carnosol, and rosmadial), among others. Some ions represented by Peaks 1, 5, and 8, could not be distinguished by their masses and fragmentation profiles; however, they showed fragment ions or losses similar to other annotations, suggesting those belong to the same class of molecules.

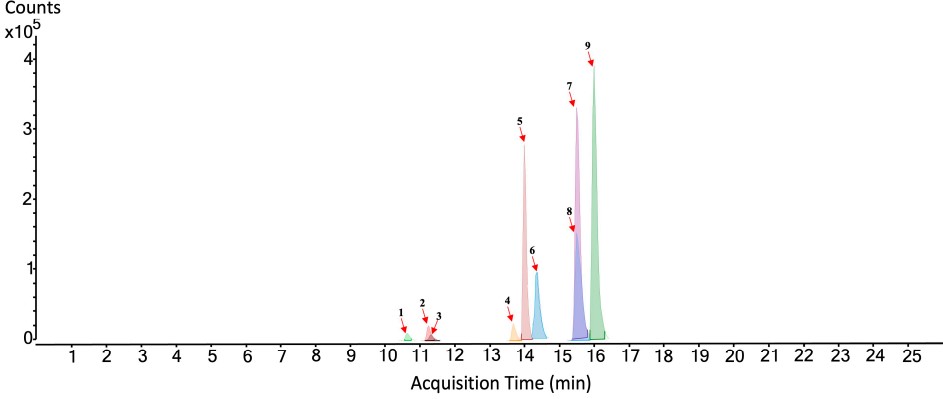

**Figure 2.** The extracted ion chromatogram (Rt: 1–25 min, EIC) for the hydro-alcoholic extract of rosemary obtained by HPLC-QTOF-MS/MS in negative ion mode.

**Table 1.** Top results of HPLC-QTOF-MS/MS analysis.

| No. Figure 2 | RT (min) | Tentative Annotation | Structure | Formula | Ion | Experimental Mass | Calculated Mass | Δ ppm |
|---|---|---|---|---|---|---|---|---|
| 1 | 10.639 | Flavonoid-glycosylated type | | $C_{22}H_{22}O_{12}$ | [M-H]− | 478.11044 | 478.111675 | −2.58 |
| 2 | 11.256 | Rosmarinic acid |  | $C_{18}H_{16}O_8$ | [M-H]− | 360.08439 | 360.085066 | −1.88 |
| 3 | 11.35 | Isorhamnetin-rutinoside |  | $C_{31}H_{28}O_{14}$ | [M-H]− | 624.14747 | 624.148454 | −1.58 |
| 4 | 13.726 | Dihydroxy-dimethoxyflavone |  | $C_{17}H_{14}O_6$ | [M-H]− | 314.07821 | 314.079587 | −4.38 |
| 5 | 14.012 | Diterpene type | | $C_{38}H_{48}O_8$ | [M+CH3COO]− | 632.33497 | 632.335467 | −0.78 |
| 6 | 14.478 | Rosmanol |  | $C_{20}H_{26}O_5$ | [M-H]− | 346.17808 | 346.178573 | −1.42 |
| 7 | 15.558 | Rosmadial |  | $C_{20}H_{24}O_5$ | [M-H]− | 344.16241 | 344.162922 | −1.49 |
| 8 | 15.597 | Diterpene type | | $C_{24}H_{26}O_9$ | [M-H]− | 458.15621 | 458.158231 | −4.41 |
| 9 | 16.077 | Carnosol |  | $C_{20}H_{26}O_4$ | [M-H]− | 330.18354 | 330.183658 | −0.36 |

Peaks 1 and 3 correspond to glycosylated flavonoids; Peak 3 was tentatively identified as isorhamnetin-rutinoside by comparison of the fragment ions (*m/z*: 315.04949 and *m/z*: 300.03166) most representative of the tandem mass spectra reported in past data for that same molecule [40]; although Peak 1 had similar fragmentation characteristics, it was not identified, but by the presence of the ion *m/z*: 315.04814, it was inferred to belong to the class of glycosylated flavonoids, well characterized in *R. officinalis* [41]. Peak 6 was tentatively noted as rosmanol because of similarities between its product ion [M-H]− 345.17019 and fragment ion $[C_{19}H_{23}O_2]-$ 283.16949 and by the information reported for this compound [42]. Interestingly, Peak 5, pending identification, presented these ions as part of its fragmentation pattern, indicating that possibly this molecule has the rosmanol structure as its core. This inference is raised, considering that *R. officinallis* uses the mevalonic acid biosynthetic pathway as part of its metabolism to produce diterpenes, which are very common in this species [41]. Peaks 7 and 8 are terpenes; Peak 7 was tentatively identified as rosmadial because of its high degree of similarity in its product ion *m/z*: 343.15454 and frag-

ment ion *m/z*: 299.16542 with that reported in previous data [43]. Peak 8 at *m/z*: 457.1481 could not be assigned a tentative name; however, the fragment ions *m/z*: 343.15461 and *m/z*: 299.16542 within its MS/MS spectrum indicated its possible relationship to rosmadial. Some diterpenes widely related to *R. officinallis*, such as carnosic acid, recognized as the main antioxidant component of rosemary, could not be identified in the extract. The absence of this component in the extract may be presumably due to two reasons: The first corresponds to the use of aged leaves, a stage where the diterpene molecule is partially consumed by non-enzymatic reactions of the diterpene into oxidized derivatives [44,45]. The second one is related to carnosic acid levels in rosemary plants exposed to stressful environmental conditions, including high temperatures and low summer rainfall, which have shown a downward trend with a concomitant increase in its main oxidation product, carnosol [46–48].

Carnosol, Peak 9, was the molecule in the group of tentatively annotated compounds with the highest number of ions detected by the mass spectrometer and was, therefore, selected as the molecule to perform virtual screening with a set of 708 human proteins.

### 3.2. Mapping Targets for Carnosol by Molecular Docking

The affinity values for carnosol on all 708 protein structures are shown in Table S2. The top 20 proteins with the best affinity values for carnosol, their Uniprot IDs, and their scored binding energies are shown in Table 2. Targets with greater binding affinity for carnosol were heat shock protein (HSP) 90-$\alpha$ (HSP 90-$\alpha$, 10.9 kcal/mol), aldo-keto reductase family 1 member C3 (AKR1C3, $-10.4$ kcal/mol), and HSP 90-$\beta$ ($-10.4$ kcal/mol).

**Table 2.** Docking scores for top 20 target proteins associated with carnosol.

| No. | PDB ID | Gene | Uniprot ID | Description | AV Binding Energy (kcal/mol) Exhaustiveness: 15 | AV Binding Energy (kcal/mol) Exhaustiveness: 32 |
|---|---|---|---|---|---|---|
| 1 | 3O0I | HSP90AA1 | P07900 | Heat shock protein HSP 90-$\alpha$ | $-10.9$ | $-10.8$ |
| 2 | 3NMQ | HSP90AB1 | P08238 | Heat shock protein HSP 90-beta | $-10.4$ | $-10.4$ |
| 3 | 1RY8 | AKR1C3 | P42330 | Aldo-keto reductase family 1 member C3 | $-10.4$ | $-10.3$ |
| 4 | 2WKM | MET | P08581 | Hepatocyte growth factor receptor | $-10.3$ | $-10.2$ |
| 5 | 1N83 | RORA | P35398 | Nuclear receptor ROR-alpha | $-10.0$ | $-10.3$ |
| 6 | 1HFQ | DHFR | P00374 | Dihydrofolate reductase (DHFR) | $-10.0$ | $-10.0$ |
| 7 | 3WHW | NUDT1 | P36639 | 7,8-dihydro-8-oxoguanine triphosphatase | $-9.9$ | $-9.8$ |
| 8 | 3L0L | RORC | P51449 | Nuclear receptor ROR-gamma | $-9.9$ | $-9.9$ |
| 9 | 2XVT | RAMP2 | O60895 | Receptor activity-modifying protein 2 | $-9.9$ | $-9.9$ |
| 10 | 1TQN | CYP3A4 | P08684 | Cytochrome P450 3A4 | $-9.8$ | $-9.8$ |
| 11 | 1DGB | CAT | P04040 | Catalase | $-9.8$ | $-9.6$ |
| 12 | 4XII | BCHE | P06276 | Cholinesterase | $-9.8$ | $-9.5$ |
| 13 | 1P8D | NR1H2 | P55055 | Oxysterols receptor LXR-beta | $-9.7$ | $-10.3$ |

| No. | PDB ID | Gene | Uniprot ID | Description | AV Binding Energy (kcal/mol) Exhaustiveness: 15 | AV Binding Energy (kcal/mol) Exhaustiveness: 32 |
|---|---|---|---|---|---|---|
| 14 | 4DRJ | FKBP4 | Q02790 | Peptidyl-prolyl cis-trans isomerase FKBP4 | −9.7 | −9.6 |
| 15 | 4AOJ | NTRK1 | P04629 | High-affinity nerve growth factor receptor | −9.6 | −9.5 |
| 16 | 1UHL | RXRB | P28702 | Retinoic acid receptor RXR-beta | −9.6 | −9.6 |
| 17 | 1ZXM | TOP2A | P11388 | DNA topoisomerase 2-alpha | −9.6 | −9.8 |
| 18 | 4FA2 | MAPK14 | Q16539 | Mitogen-activated protein kinase 14 | −9.5 | −9.3 |
| 19 | 4J52 | PLK1 | P53350 | Serine/threonine-protein kinase PLK1 | −9.5 | −9.5 |
| 20 | 3X36 | VDR | P11473 | Vitamin D3 receptor | −9.2 | −9.2 |

*3.3. Mapping Targets of Carnosol by Comparative Toxicogenomics Database (CTD)*

The potential targets for carnosol mapped in CTD with the keyword "Carnosol" are shown in Table 3. We found 33 targets related to carnosol.

**Table 3.** Potential targets for carnosol mapped in CTD.

| No. | PDB ID | Gene | Uniprot ID | Description |
|---|---|---|---|---|
| 1 | 1TNR | TNF | P19438 | Tumor necrosis factor |
| 2 | 5UCX | PRDX3 | P30048 | Peroxiredoxin 3 |
| 3 | 2VGE | RELA | Q8WUF5 | RELA proto-oncogene, NF-kB subunit |
| 4 | 6EHA | HMOX1 | P09601 | Heme oxygenase 1 |
| 5 | 7O7B | NFE2L2 | Q16236 | NFE2-like bZIP transcription factor 2 |
| 6 | 3E2M | ICAM1 | P20701 | Intercellular adhesion molecule 1 |
| 7 | 4Q7H | IFNG | | Interferon gamma |
| 8 | 6ZZU | TH | P07101 | Tyrosine hydroxylase |
| 9 | 4S0O | BAX | Q07812 | BCL2-associated X, apoptosis regulator |
| 10 | 5UUK | BCL2 | Q16548 | BCL2 apoptosis regulator |
| 11 | 4H1V | DNM1L | O00429 | Dynamin 1-like |
| 12 | 1NZN | FIS1 | Q9Y3D6 | Fission, mitochondrial 1 |
| 13 | 6OYW | MAP3K5 | Q99683 | Mitogen-activated protein kinase kinase kinase 5 |
| 14 | 4G1W | MAPK8 | P45983 | Mitogen-activated protein kinase 8 |
| 15 | 7CML | MAPK9 | P45984 | Mitogen-activated protein kinase 9 |
| 16 | 6JFL | MFN2 | O95140 | Mitofusin 2 |
| 17 | 7KKM | PARP1 | O95271 | Poly(ADP-ribose) polymerase 1 |
| 18 | 7RNF | CASP3 | P42574 | Caspase 3 |
| 19 | 5M8R | TYR | P17643 | Tyrosinase |
| 20 | 5NJ8 | AHR | P35869 | Aryl hydrocarbon receptor |

**Table 3.** *Cont.*

| No. | PDB ID | Gene | Uniprot ID | Description |
|-----|--------|------|-----------|-------------|
| 21 | 2HI4 | CYP1A2 | P05177 | Cytochrome P450 family 1 subfamily A member 2 |
| 22 | 4GQS | CYP2C19 | P33261 | Cytochrome P450 family 2 subfamily C member 19 |
| 23 | 6VLT | CYP2C9 | P11712 | Cytochrome P450 family 2 subfamily C member 9 |
| 24 | 4XRY | CYP2D6 | P10635 | Cytochrome P450 family 2 subfamily D member 6 |
| 25 | 4D7D | CYP3A4 | P08684 | Cytochrome P450 family 3 subfamily A member 4 |
| 26 | - | GCLC | | Glutamate–cysteine ligase catalytic subunit |
| 27 | - | GCLM | | Glutamate–cysteine ligase modifier subunit |
| 28 | 4KIK | IKBKB | O14920 | Inhibitor of nuclear factor kappa B kinase subunit beta |
| 29 | 1A02 | JUN | P05412 | Jun proto-oncogene, AP-1 transcription factor subunit |
| 30 | 6G54 | MAPK1 | P28482 | Mitogen-activated protein kinase 1 |
| 31 | 6GES | MAPK3 | P27361 | Mitogen-activated protein kinase 3 |
| 32 | 6Y1J | NFKBIA | P25963 | NFKB inhibitor alpha |
| 33 | 5YD6 | NR4A2 | P43354 | Nuclear receptor subfamily 4 group A member 2 |

All targets for carnosol identified with docking and CTD were used for further analysis.

### 3.4. Molecular Docking Validation

In order to assess the prediction capability of the docking protocol used in this study, a series of re-docking experiments were carried out with 13 target proteins associated with carnosol. The results are presented in Table S3. Most X-ray complexes showed similar conformational similarity with those obtained in this study, with RMSD values that varied from 0.0002 to 2.387 Å, with an average of 0.867 Å.

### 3.5. Analysis of Pharmacological Network

The analysis of the pharmacological network was carried out in order to elucidate the relationships among targets as well as their corresponding biochemical pathways targeted by studied ligand. The protein–protein interaction network (Figure 3) contains 50 nodes and 594 edges, the average degree value of the node is 11.8, the average Betweenness Centrality is 0.52, and the average Closeness Centrality is 0.52. Overall, there are eight nodes with degree value, Betweenness Centrality, and Closeness Centrality greater than the average. These may be the main potential targets for carnosol to exert its function (Table 4).

**Table 4.** Key protein topological parameters of protein interaction network.

| Name | Degree | Betweenness Centrality | Closeness Centrality |
|------|--------|------------------------|---------------------|
| TNF | 30 | 3.4008 | 0.7101 |
| CAT | 29 | 3.0840 | 0.6901 |
| JUN | 29 | 1.4134 | 0.6901 |
| HSP90AA1 | 29 | 2.2865 | 0.6901 |
| CASP3 | 29 | 1.5676 | 0.6622 |
| MAPK3 | 26 | 1.4197 | 0.6364 |
| MAPK1 | 22 | 1.0351 | 0.6050 |
| AHR | 18 | 3.2344 | 0.6050 |

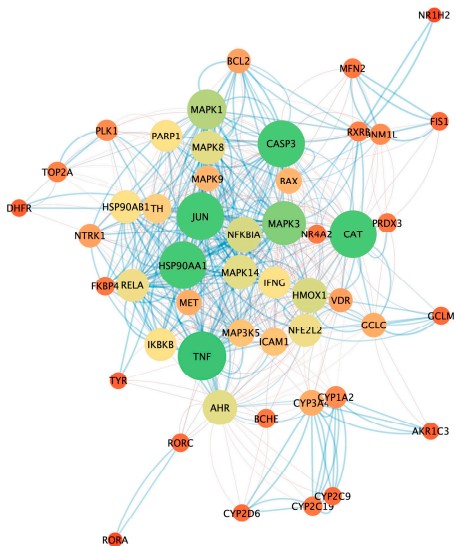

**Figure 3.** Pharmacological network for carnosol. The size and color of the circle varied with the degree value. The width and color of edges varied with the combined score.

### 3.6. GO and KEGG Pathway Enrichment Analysis

To analyze the set of targets for carnosol, we classified them according to the term GO and KEGG pathways using by WebGestalt tool. GO enrichment analysis for carnosol-modulated targets (Figure 4) showed the most significant biological process (BP), molecular function (MF), and cellular component (CC). BP enrichment contained mainly genes involved in the following: metabolic process (47/47), response to stimulus (47/47), biological regulation (44/47), developmental process (41/47), multicellular organismal process (40/47), cell communication (38/47), cellular component organization (33/47), localization (27/47), multi-organism process (24/47), cell proliferation (22/47), reproduction (14/47), and growth (6/47). MF enrichment contained mainly the following target genes: protein binding (43/47) and ion binding (34/47). KEGG pathway enrichment analysis (Figure 5) showed that 33 potential target genes were enriched, and 12 signal pathways were significantly related with the target genes (FDR ≤ 0.05).

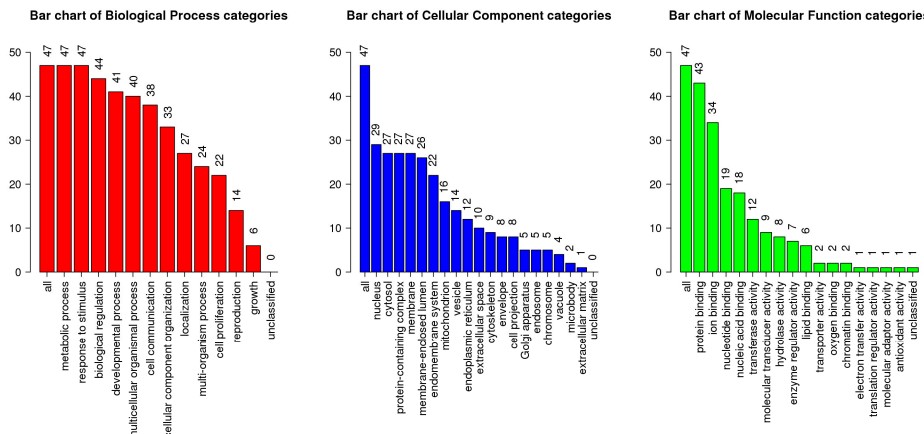

**Figure 4.** GO enrichment analysis of the identified targets to carnosol in terms of biological process (red), cellular component (violet), and molecular function (green). The bar chart shows the number of targets in each term.

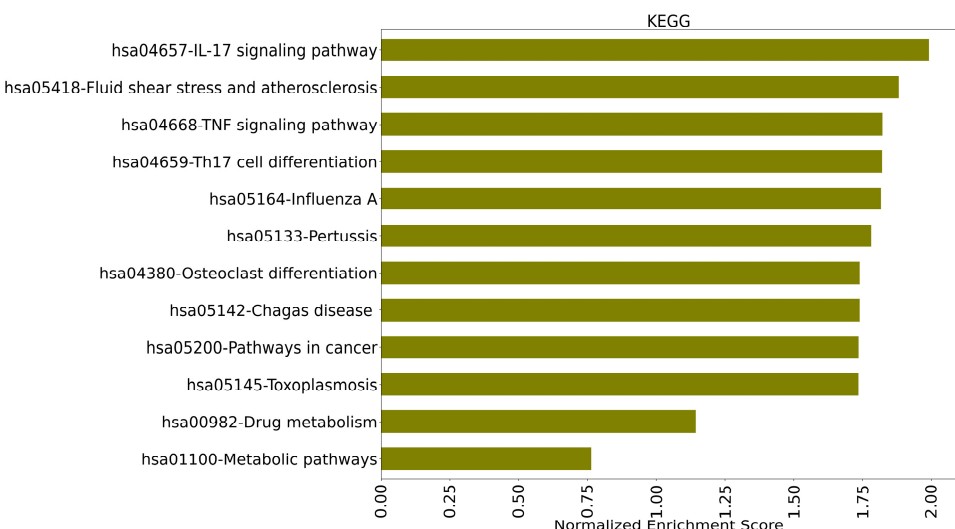

**Figure 5.** Top 12 pathways associated with carnosol targets according to the KEGG pathways database. The normalized enrichment score showed the enrichment degree in KEGG.

### 3.7. Chemical Compound–Target–Pathway

The target–pathway network is constructed to elucidate the interactions among carnosol, targets and pathways. The target–pathway network associated with carnosol (Figure 6) shows 48 nodes and 159 edges. Orange circles correspond to targets, and green triangles represent pathways.

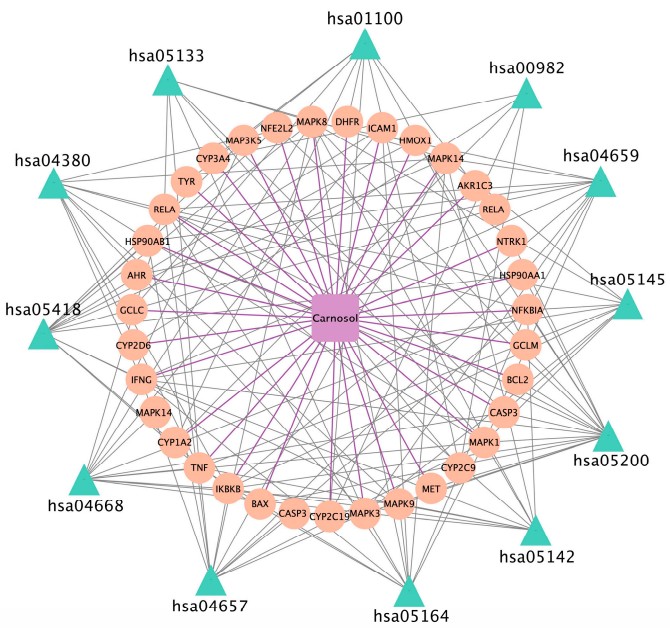

**Figure 6.** Compound (Carnosol)–target–pathway network.

### 3.8. Molecular Docking Results

Docking results for the eight targets with degree value, Betweenness Centrality, and Closeness Centrality above the average are shown in Table 5, and the top three complexes are shown in Figure 7. Considering the lowest affinity energy as the best interaction possibility, HSP90AA1, MAPK1, and MAPK3 proteins have the highest potential to be carnosol targets. Carnosol binding to HSP90 is supported by the hydrophobic interactions with LEU107A, PHE138A, VAL150A, TRP162A, and the hydrogen bonds with ASN51A, TYR139A, and PHE138A (Figure 7A). The binding of carnosol to MAPK1 is supported

by the hydrophobic interactions with ILE31A, LYS54A, ILE84A, GLN105A, LYS114A, and LEU156A, and the hydrogen bonds with LYS54A and ASP111A (Figure 7B). Carnosol was coupled to MAPK3 through hydrophobic interactions with TYR53A, LYS71A, ILE73A, and LEU187A, and the hydrogen bonds with TYR53A and ASP184A (Figure 7C).

**Table 5.** Main potential molecular docking binding energy for target–carnosol complexes.

| No. | PDB ID | Gene | Ref | Compound | AV Binding Energy (kcal/mol) | Interactions |
|---|---|---|---|---|---|---|
| 1 | 3O0I | HSP90AA1 | [49] | Carnosol | −10.8 | LEU107A, PHE138A, VAL150A, TRP162A, ASN51A, TYR139A |
| 2 | 1G54 | MAPK1 | [50] | | −8.5 | ILE31A, LYS54A, ILE84A, GLN105A, LYS114A, LEU156A, ASP111A |
| 3 | 6GES | MAPK3 | [51] | | −8.3 | TYR53A, LYS71A, ILE73A, LEU187A, ASP184A, LYS168A |
| 4 | 1DGB | CAT | [52] | | −7.9 | PHE198A, VAL302A, ALA445A, PHE446A, VAL450A, HIS305A |
| 5 | 1A02 | JUN | [53] | | −7.6 | ARG541N, PRO566N, GLN669N ASP464N |
| 6 | 5NJ8 | AHR | [54] | | −7.5 | TYR76A, TYR137A, LYS80A |
| 7 | 7RNF | CASP3 | [55] | | −7.3 | PHE247D, PHE250D, ASN208D, GLU246D, PHE247D |
| 8 | 1TNR | TNF | [56] | | −6.3 | ALA30A, PHE53A, PHE169A, LEU171A, ALA170A |

### 3.9. Molecular Dynamic (MD) Simulation

The MD simulations of carnosol complexed with HSP90 and MAPK1 were investigated during a simulation time of 100 ns to analyze the flexibility and stability of the complex over time. The stability of HSP90 and MAPK1 proteins with their respective co-crystallized ligands was also simulated under the same conditions in order to compare the existing conformational and energetic changes through calculations of RMSD, RMSF, and MMGBSA (Figure 8).

#### 3.9.1. Root Mean Square Deviations (RSMD)

The RMSD results showed that the backbone atoms of HSP90 in complex with the co-crystallized ligand (P54) or carnosol underwent few fluctuations in their conformational structure. The minimum and maximum RMSD values were 0.09 and 0.21 nm for HSP90-P54 and 0.08 and 0.22 nm for HSP90-carnosol, respectively (Figure 8(A1)). After ∼50 ns of simulation, the RMSD of HSP90 with both ligands fluctuated between ∼0.13 and ∼0.20 nm for P54 and ∼0.13 and ∼0.22 nm for carnosol, reaching a metastable state in the middle of the simulation period. These low RMSD values suggest the protein achieved high conformational stability with both ligands.

The RMSD profile of the complex MAPK1-carnosol or MAPK1-6H3 (co-crystallized ligand) displayed minimum and maximum RMSD values of 0.18 and 5.14 nm for MAPK1-6H3 and 0.31 and 6.91 nm for MAPK1-carnosol, respectively (Figure 8(B1)). During the first 30 ns, the interactions in both complexes depicted slight fluctuations with an RMSD of less than 3 nm. After 30 s, the MAPK1-carnosol complex increased some nm compared with MAPK1-6H3. However, after 70 ns of simulation, the MAPK1-carnosol complex deviated markedly from the co-crystallized ligand, and both systems reached some equilibrium after 95 ns producing a stable trajectory. After the simulation period, the MAPK1-6H3 and the MAPK1-carnosol complexes had mean RMSD values of 2.70 and 3.75 nm, respectively.

The RMSD plots for MAPK1 showed that the systems did not reach equilibrium; therefore, the radius of gyration (Rg) was determined for MAPK1 complexes. The Rg

is a physical quantity that describes the compactness of the protein structure, that is, a low value of Rg describes a more rigid structure during the simulation. The MAPK1-6H3 complex (Figure S2A) maintained the equilibrium, with an average Rg value of 1.7 nm during the first 50,000 ps (50 ns). From 50,000 ps to 80,000 ps, there were fluctuations between 1.7 and 1.9 nm, and after 80,000 ps, the Rg decreased to 1.7 nm. The protein without the ligand maintained an average Rg of 1.7 nm throughout the simulation. The MAPK1-carnosol complex (Figure S2B) maintained the equilibrium, with an average Rg value of 2.15 nm during the first 60,000 ps (60 ns), and from 60,000 to 95,000 ps, there were fluctuations between 2.15 and 2.4 nm; then, the Rg decreased to 2.15 nm, whereas the protein without the ligand maintained an average Rg of 2.15 nm throughout the simulation. Both complexes showed a large variation compared with the unliganded protein, the greatest being observed with carnosol (Figure S2).

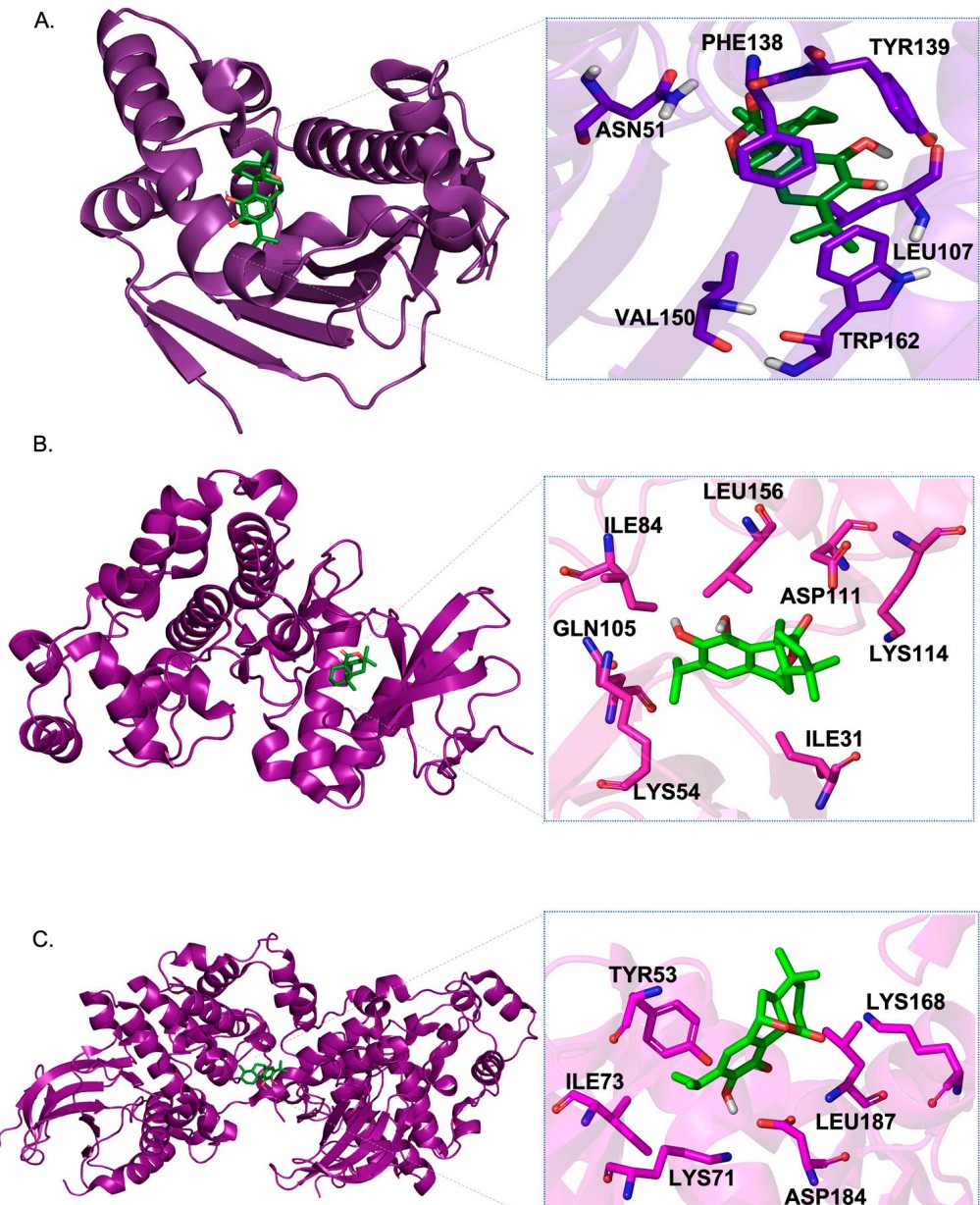

**Figure 7.** 3D structures of the three best docking results. (**A**) HSP90 and carnosol; (**B**) MAPK1 and carnosol; and (**C**) MAPK3 and carnosol.

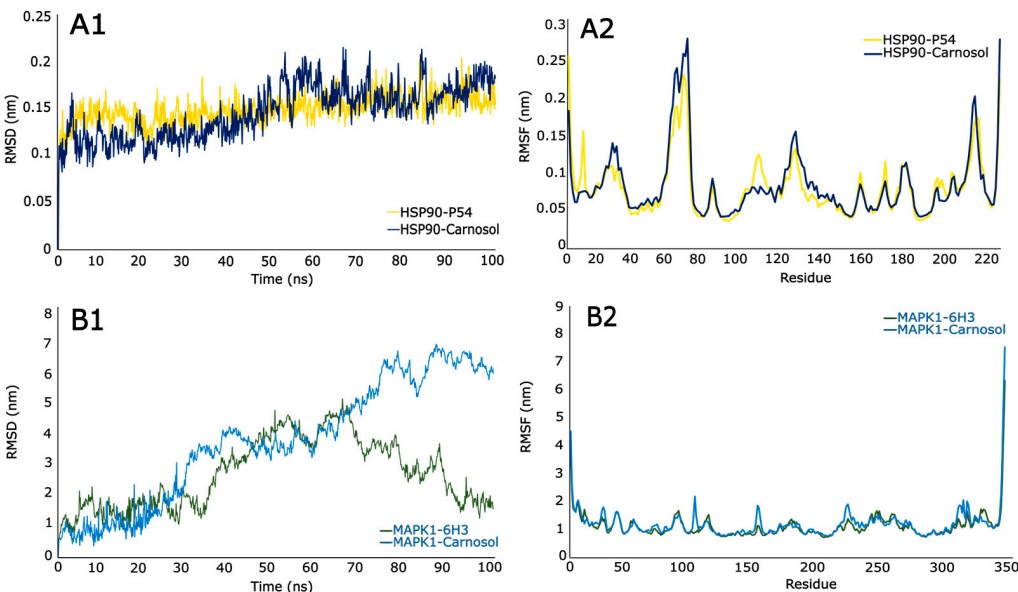

**Figure 8.** Backbone RMSD plots of ligand native and carnosol with HPS90 (**A1**) and MAPK1 (**B1**); and RMSF plots of ligand native and carnosol with HPS90 (**A2**) and MAPK1 (**B2**).

### 3.9.2. Root Mean Square Fluctuations (RSMF)

The RSMF profile for HSP90 with both ligands (Figure 8(A2)) showed greater fluctuations at the ends of the protein and in the stretch between the ~65 and ~76 amino acids, indicating that in this group of residues, there is greater flexibility, and therefore, greater potential to interact with both ligands. This flexibility can be attributed to the beta turn motifs that form the residue groups, ranging from 65 to 76 and from 210 to 220 in HSP90.

Both ligands behaved similarly; however, certain fluctuations were observed between residues 24–26 and 107–122. Between amino acid residues 107–122 of HSP90-P54, a slight difference was observed with respect to HSP90-carnosol binding, which may be attributed to the additional hydrophobic interactions that carnosol establishes with those amino acids.

The RMSF results for MAPK1 with 6H3 and carnosol (Figure 8(B2)) showed that the fluctuations were not so significant; however, one of the stretches with the highest fluctuations was between residues 94 and 96, which are part of the active site of MAPK1 where the co-crystallized ligand binds. In addition, other peaks with above-average fluctuations, other than the residues involved in the interactions with 6H3 and Carnosol, were also observed, suggesting other interaction possibilities. For example, amino acids ASP109, and GLY231 presented one of the highest peaks of the RMSF profile in carnosol, indicating that these residues have flexibility when binding to Carnosol, even more than when binding to the co-crystallized ligand. This flexibility may be related to the beta turn motif present in these stretches of the MAPK1 secondary structure.

### 3.9.3. Molecular Mechanics Energies Combined with Surface Area Continuum Solvation (MMGBSA)

MMGBSA calculations in molecular dynamics allowed estimation of the total binding free energy of HSP90 complexes with P54 and carnosol and of MAPK1 with 6H3 and carnosol (Table 6). A positive value of total binding free energy suggests unfavorable binding, and the more negative, the more favorable the binding. Within the complexes studied, HSP90 has a more favorable binding free energy with carnosol (−28.4237 kcal/mol) than with the co-crystallized ligand P54 (−6.1978 kcal/mol), as does MAPK1 with carnosol (−27.7457 kcal/mol) compared with its co-crystallized 6H3 (−27.6204 kcal/mol), indicating that both proteins have a thermodynamically more stable association with carnosol than with its co-crystallized ligand, the most significant difference being in the HSP90-P54 complex.

**Table 6.** MMGBSA-based total binding free energies along with standard deviation.

| Complex | Total Binding Free Energy (kcal/mol) | Standard Deviation |
|---|---|---|
| HSP90-P54 | −6.1978 | 2.8193 |
| HSP90-Carnosol | −28.4237 | 2.9445 |
| MAPK1-6H3 | −27.6204 | 2.7037 |
| MAPK1-Carnosol | −27.7457 | 3.7537 |

## 4. Discussion

In the current study, the plausible human targets and the underlying biochemical pathways intervened by carnosol were analyzed. We used two ways for mapping candidate human targets for carnosol, an in silico prediction strategy (20 targets found) and a database search (33 targets found). Finding 53 targets made it possible to construct intricate networks for proteins and pathways modulated by carnosol, showing the possible biological processes through which this molecule achieves its pharmacological effect.

Key protein topological parameters of protein interaction (degree value, Betweenness Centrality, and Closeness Centrality) of eight nodes were above average. These nodes, chaperone (HSP90AA1) and catalase (CAT), both mapped with in silico screening, and kinases (MAPK1 and MAPK3), cytokine (TNF), caspase (CASP3), and transcription factors (AHR and JUN), mapped with CTD analysis, participated in the process of immune-inflammatory responses [57–60]. HSP90AA1 and MAPK1 are the top two potential carnosol targets according to the exploratory docking and network topology calculations; however, the binding energy calculations suggest that carnosol had a stronger binding to HSP90 than MAPK1. These results were validated with the stability shown by the HSP90-carnosol complex during the 100 ns of molecular dynamics simulation, whereas the MAPK1-carnosol complex did not reach equilibrium in the same period. HSP90 has been previously reported as a target for carnosol. For example, carnosol inhibited NLRP3 inflammasome activation by directly targeting Hsp90 and blocking its ATPase activity in female mice with lipopolysaccharide (LPS)-induced septic shock [61]. In HaCaT and MSK-Leuk1 cells, established from a premalignant dysplastic leukoplakia lesion, carnosol inhibited Hsp-90 activity, causing rapid decrease in AhR levels and then suppression of the induction of Cytochrome P450 1A1 (*CYP1A1*) and Cytochrome P450 1B1 (*CYP1B1*) [62].

MET is another potential binding protein for carnosol identified by docking calculations in this study, also reported in the literature as a carnosol target. MET is a receptor that translates proliferation and survival cell signals by binding to the hepatocyte growth factor ligand, HGF [63]. Signaling cascades activated following MET/HGF interaction include RAS-ERK and PI3 kinase-AKT [64]. The aberrant signaling of both pathways occurs in most types of cancer; thus, it is under intense research to identify new inhibitors of their associated proteins [65,66]. Carnosol has been shown to be a potential MET inhibitor in pancreatic cancer stem cell subpopulations by targeting the kinase domain of MET, and thereby suppressing downstream AKT survival, attenuating cancer cell growth and motility [67].

The effect of carnosol binding to some targets identified by docking calculations in this study have not been clearly explored in the literature; for example, the carnosol binding to AKR1C3 [68]. AKR1C3 is an Aldo-keto reductase protein involved in cell proliferation and endocrine disorders associated with prostate and breast cancer [69]. In prostate cancer, AKR1C3 acts through two mechanisms. In the first, AKR1C3 catalyzes the synthesis of the formation of the potent androgens, testosterone (T) and 5-dihydrotestosterone (5-DHT) that bind to the androgen receptor and promote cell growth [70]. AKR1C3 also catalyzes the conversion of PGH2 and PGD2 to prostaglandin (PG) F2 and 11-PGF2, molecules that promote tumor cell proliferation [70]. Although rosemary extract has showed significant inhibition of proliferation, survival, and migration of PC-3 prostate cancer cells by targeting

Akt and mTOR [71], the anti-proliferative role that carnosol may play on prostate cancer through AKR1C3 has not been explored.

The actions and interactions of the multiple targets targeted by carnosol were interpreted using systems-biology-based network pharmacology. We carried out GO and KEGG pathways analysis for carnosol targets through WebGestalt to channel information from the pharmacology network. This analysis showed that carnosol is closely related to infection (Chagas disease, influenza A, toxoplasmosis, and pertussis) and inflammation (IL-17 and TNF signaling pathway and Th-17 cell differentiation). Some of the pathways modulated by carnosol are related to inflammatory effects produced by in infectious diseases; for example, Th17 cell differentiation is a pathway which plays an important role in the inflammatory response associated with antigen-presenting cells (APCs) for recognition by certain T lymphocytes [72,73]. This pathway was targeted by carnosol through TNF, CASP3, HSP90AA1, MAPK3, MAPK1, NFKBIA, MAPK14, MAPK8, RELA, HSP90AB1, IFNG, and IKBKB [60]. These findings may help explain the use of rosemary as a treatment for inflammatory respiratory diseases, such as asthma [74], and infectious diseases, such as COVID-19 [75], labial herpes [76], and candidiasis [77], among others.

This constellation of findings suggests that carnosol, the most ionized molecule in alcoholic-water extract of *R. officinalis* by targeting key targets, may induce an immuno-inflammatory response, inhibiting excessive immune response and a storm of inflammatory factors. These GO terms and pathways findings are supported by other studies that show the suppressive effect of carnosol on fibrosis, oxidative stress, and inflammation in mouse lungs [78] and its antimicrobial activity against bacteria and yeasts with dermatological relevance [79].

## 5. Conclusions

Computational chemistry and bioinformatics analysis were carried out to identify the potential pharmacological mechanisms of carnosol, elucidating their targets and related pathways. This small molecule has the capacity to target various proteins, such as HSP90AA1, MAPK1, MAPK3, CAT, JUN, AHR, CASP3, and TNF, and is able to modulate immune-inflammatory pathways, including IL-17 and TNF signaling, Th-17 cell differentiation, chagas, influenza A, toxoplasmosis, and pertussis diseases. This information can be used as a basis to elucidate the processes modulated by the ingestion of R. officinalis, in particular by carnosol, as a therapeutic strategy to combat infection and inflammation.

**Supplementary Materials:** The following supporting information can be downloaded at: https://www.mdpi.com/article/10.3390/scipharm91020019/s1, Figure S1: Compound fragment spectrum results (MS/MS); Figure S2: Radius of gyration for MAPK1 complexes. Radius of gyration for MAPK1-6H3 complex (A) and radius of gyration for MAPK1-carnosol complex (B); Table S1: Fragment ions of chemical constituents tentatively identified in Rosmarinus officinalis by HPLC-QTOF-MS/MS; Table S2: Docking score of 708 human proteins with carnosol; Table S3: RMSD and Superposition between the crystallographic structures of 13 the complexes and the resultant docking pose.

**Author Contributions:** Conceptualization, M.T.-A. and J.O.-V.; methodology, D.P.-V., M.T.-A. and M.A.-M.; writing—review and editing, M.T.-A., J.O.-V., A.C., K.C.-G. and E.I.; supervision K.C.-G., J.O.-V. and E.E.S.; project administration K.C.-G., J.O.-V. and E.E.S.; funding acquisition K.C.-G., J.O.-V. and E.E.S. All authors have read and agreed to the published version of the manuscript.

**Funding:** This research was funded by the Ministry of Science, Technology and Innovation (Minciencias), the Ministry of Education, the Ministry of Industry, Commerce and Tourism, and ICE-TEX, Program Ecosistema Científico–Colombia Científica, from the Francisco José de Caldas Fund (Grant RC-FP44842-212-2018), Colombia. Minciencias, Sistema General de Regalías de Colombia, National Program for Doctoral Formation (BPIN 2020000100364, State of Sucre and University of Cartagena, 2020). The funders had no role in the design of the study, the collection, analyses, or interpretation of data, the writing of the manuscript or the decision to publish the results.

**Institutional Review Board Statement:** Not applicable.

**Informed Consent Statement:** Not applicable.

**Data Availability Statement:** Not applicable.

**Acknowledgments:** Initiative to support Research Groups and Doctoral Programs. Vice-Presidency for Research. University of Cartagena, Cartagena, Colombia (2020–2023).

**Conflicts of Interest:** The authors declare no conflict of interest.

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
