# Peer review of "Mapping Protein Targets of Carnosol, a Molecule Identified in Rosmarinus officinalis: In Silico Docking Studies and Network Pharmacology"

_scipharm, doi:10.3390/scipharm91020019_

Round 1

Reviewer 1 Report

I the work titled "Mapping of protein targets of carnosol, a molecule identified in Rosmarinus officinalis: in silico docking studies and network pharmacology" Needs to be improved to increase the overall scientific merit of the reseach work.

Please find attached the comments.

Author Response

Answer to Reviewers

  1. Introduction, line 34, please look into the work “specie” and replace with the right term.

Answer

The term “specie” was changed to “Rosemary” (Line 36)

  1. Is the method for plant material extraction, a novel one? If no, then please cite any previous papers for comparison of protocol.

Answer

The extraction method of the plant is not new, it has been reported previously in Duran-Izquierdo et al., 2022. Line 78

  1. Similarly, please provide protocol citations for “Sample preparation and HPLC-QTOF-MS/MS conditions” section.

Answer

The sample preparation protocol and HPLC-QTOF-MS/MS conditions have been previously reported in Duran-Izquierdo et al., 2022. Line 110

  1. Please provide more details on the protein optimizing parameters using “SYBYL-X 2.0”

Answer

The methodology for protein preparation has been standardized by the research group and used with the details described from line 114 to 123 in previous references [Cabarcas-Montalvo et al., 2016; Coronado‐Posada et al., 2021]

  1. Citations missing for PDB database, the protein structure in PDB (13 structures shortlisted for molecular docking) and its corresponding publication.

Answer

Table S3 shows the PDB code, the native ligand name, its corresponding reference and binding affinity, as well as the RMSD comparing the native ligand binding site with carnosol binding site.

  1. The PubChem entry for Carnosol needs to be cited too along with PubChem database. Citations for both are missing. Overall lot of citations are misssing.

Answer

The reference associated with the reuse of the 3D structure image of carnosol can be seen on lines 125 and 126.  The general reference of PubChem data and the individual carnosol record are shown on lines 127 and 128.

  1. Gaussian tool was used for optimization of Carnosol. To the best of my knowledge, it is a commercially available tool. Please mention and version and comment of the availability of license to use the same.

Answer

The version used was Gaussian 09 Unix Single Machine Type Site License for AMD64 Linux (Gaussian, Inc., Wallingford CT, 2009) acquired in 2011 by Environmental and Computational Chemistry Group, University of Cartagena-Colombia (see line 128).

  1. In general terms, exhaustiveness of 15 is not a very robust value. Please justify this either by citing other work or enhance it by improving the value.

Answer

The use of exhaustiveness of 15 is supported by the literature (Fedorov et al., 2015, see line 133). In addition to that, the top 20 of the carnosol targets was docketed once again but using exhaustiveness of 32, which has been previously used in the work of Eberhardt et al., 2021 (See line 134). The results of the docking with exhaustiveness of 32 for the top 20 are shown in the last column of Table 2.

  1. Please look into other online options like PLIP (https://plip-tool.biotec.tu-dresden.de/plipweb/plip/index) for mapping protein ligand and it will assist in better understanding of interaction pattern.

Answer

Following your recommendations, PLIP (https://plip-tool.biotec.tu-dresden.de/plipweb/plip/index) was used to map the protein residues involved in carnosol binding (See lines 157 - 159). The results are shown in the last column of Table 5 and in Figure 7.

  1. “The pathways obtained with KEGG were marked in the bar chart using python”. Very generic statement, please provide more details on which package and cite it.

Answer

The pathways obtained with KEGG using the WebGestalt tool were visualized in the bar chart using matplotlib library in python (Hunter, 2007). See line 172 and 173.

  1. Docking scores for top 20 proteins is provided in the table. I did not find the S2 table with the complete list for review? Wondering if its actually missing?

Answer

Docking scores for 708 human proteins with carnosol are shown in Table S2 of the supplementary material.

  1. In line 290 and 291, please correct it to (–7 kcal/mol).

Answer

The statement was modified to take into account the reviewers' suggestions

  1. Please elaborate on how from 703 PDB structures where the authors able to shortlist the 13 structures for molecular docking? Some clarity is missing. Too much of protein numbers are given throughout the methods and results section

Answer

In total, there were 708 PDB structures docked with carnosol (See the line 250). Of the 708 (Table S2 - Supplementary material), the best 20 were selected to be shown in Table 2 (See the line 251). These dockings were validated by calculating the RMSD between a pose of carnosol docked with respect to original crystallographically of 13 proteins randomly chosen from the total of docked proteins (Table S2 - Supplementary material) (See the line 271 to 272).

  1. “Although there is evidence about carnosol targets, 317 there is not information about network pharmacology of carnosol to understand its actions and interactions with multiple targets.” This is very confusing statement because if target is already known then the need for pharmacology- based network is not needed.

Answer

Taking into account that carnosol, as well as many other therapeutics, targets multiple proteins rather than single targets, this paper attempts to address two objectives, the first one is to expand the list of proteins known in the literature to be targeted by carnosol, taking advantage of the bank of 708 docketed human proteins related to different biological processes, which was created by the research group.  The second objective is to understand the actions and interactions of the multiple targets targeted by carnosol and thereby the related mechanisms of action using network pharmacology, which is based on systems biology and polypharmacology, to offer a novel network mode of "multiple targets, multiple effects, complex diseases" and replaces the rational "magic bullet" drug design with the multi-target "magic shotgun" approach (Zhang, et al., 2013).

Due to the confusion associated with the previous statement, the authors changed it to a new statement “Some of the targets reported for carnosol in the literature also showed a high possibility of interaction through exploratory docking and network topology calculations in this study” (Line 377-379).

  1. In continuation, molecular docking followed by MD simulations of up to 100 ns was the best approach for the paper to prove the mode of action.

Answer

Following the recommendations of the reviewers, a 100 ns molecular dynamics simulation was carried out for the two best complexes. The methodology is described between lines 186-201 and the results between lines 331 and 391.

  1. In silico research papers are not complete without simulation results, please perform MD simulation for top complexes. I would request to authors to look into above mentioned comments to overall improve the scientific merit of the work done.

Answer

See answer to question 15.

Reviewer 2 Report

The manuscript “Mapping of protein targets of carnosol, a molecule identified in Rosmarinus officinalis: in silico docking studies and network pharmacology” by Maria Taboada-Alquerque et al. has applied molecular docking to screen plausible human targets for carnosol, which appears to be a starting point for future in-depth in vitro and in silico studies. Pathway analyses were also performed to understand the pharmacological effects of carnosol. I think this manuscript merits publication in Sci. Pharm. But the following concerns should be well addressed before it becomes publishable.

1. About HPLC-QTOF-MS/MS:

The authors state that “Carnosol, peak 9, was the molecule in the group of tentatively annotated compounds with the highest number of ions detected by the mass spectrometer and was therefore selected as a precursor to perform virtual screening with a set of 707 human proteins” (page6, lines 215–217), which confuses me as follows:

1) How did the authors attribute the peak 9 to carnosol? How are “compounds with the highest number of ions” related to carnosol? Or am I missing anything here?

2) The authors “selected (carnosol) as a precursor to perform virtual screening”. Do the authors mean the molecule employed in the docking is a derivative of carnosol by saying “precursor”?

2. About molecular docking:

1) Carnosol was docked into 708 human proteins. What is the rationale for selecting these 708 human proteins? I don’t see an explanation anywhere.

2) Potential targets were identified based on an affinity cutoff of -10.4 kcal/mol. Why was this cutoff chosen? I don’t see a justification anywhere.

3. About pathway analyses:

1) The pathway analyses further screened 8 human proteins subjected to docking. Why do the new docking affinities in Table 5 diff the old values in Table 3, assuming the same protein structures and docking protocol were used? In addition, I don’t know what the authors are trying to convey by saying that “the lowest affinity energy between carnosol and proteins as the best possibility of interaction (<7.0 Kcal/mol)” (page 12, lines 289–291).

2) In Fig. 7, the authors display the binding modes of carnosol to HSP90, MAPK1, and MAPK3, whose affinities respectively rank 1, 3, and 4 in Table 5. Then what about the binding to CAT, which has the 2nd highest affinity?

4. Other comments:

1) Where are the supporting figures and tables? I don’t see them anywhere.

2) Table 3: the 2nd column should be “PDB ID”, the 3rd column should be “Gene”, the “Uniprot ID” information is missing.

Author Response

The manuscript “Mapping of protein targets of carnosol, a molecule identified in Rosmarinus officinalis: in silico docking studies and network pharmacology” by Maria Taboada-Alquerque et al. has applied molecular docking to screen plausible human targets for carnosol, which appears to be a starting point for future in-depth in vitro and in silico studies. Pathway analyses were also performed to understand the pharmacological effects of carnosol. I think this manuscript merits publication in Sci. Pharm. But the following concerns should be well addressed before it becomes publishable.

Thanks for the comment

  1. About HPLC-QTOF-MS/MS:

The authors state that “Carnosol, peak 9, was the molecule in the group of tentatively annotated compounds with the highest number of ions detected by the mass spectrometer and was therefore selected as a precursor to perform virtual screening with a set of 707 human proteins” (page6, lines 215–217), which confuses me as follows:

1) How did the authors attribute the peak 9 to carnosol? How are “compounds with the highest number of ions” related to carnosol? Or am I missing anything here?

Answer

The analysis of the rosemary extract using a liquid chromatograph coupled to a mass spectrometer showed many chemical characteristics. These chemical characteristics could be primary or secondary metabolites, and noise or artefacts, however we focused on secondary metabolites for their biological activities. Of the secondary metabolites detected by the analysis, only 9 were annotated for their similarity in exact mass and mass spectra to the information reported for these molecules in the literature (See Table S1, and the mass spectrum for all compounds). Of the 9 metabolites detected, carnosol was the molecular ion with the highest abundance according to the total ion chromatogram or TIC, which is the representation of everything that was detected during the analysis. The ions of interest, in our case the 9 ions corresponding to the annotated molecules, were extracted from the TIC and they are plotted in the extracted ion chromatogram (EIC) (Fig. 2).

2) The authors “selected (carnosol) as a precursor to perform virtual screening”. Do the authors mean the molecule employed in the docking is a derivative of carnosol by saying “precursor”?

Answer

No, the molecule used to present the virtual screening was carnosol. The meaning of precursor is related to the precursor ion of m/z: 329.17600 corresponding to carnosol, detected during the mass spectrometry analysis. Due the confusion, the authors decided to modify the term "precursor" to "molecule". See line 242.

  1. About molecular docking:

1) Carnosol was docked into 708 human proteins. What is the rationale for selecting these 708 human proteins? I don’t see an explanation anywhere.

Answer

These 708 proteins were selected because the environmental and computational chemistry research group of the University of Cartagena created an internal database of optimized proteins involved in different biological processes, for research purposes.

2) Potential targets were identified based on an affinity cutoff of -10.4 kcal/mol. Why was this cutoff chosen? I don’t see a justification anywhere.

Answer

-10.4 kcal/mol is not a cutoff, the purpose of including this value in the statement was to highlight that the three best affinity values were below -10.4 kcal/mol. Due to the confusion, the authors removed the value of (<-10.4 kcal/mol) from the statement. See line 252

  1. About pathway analyses:

1) The pathway analyses further screened 8 human proteins subjected to docking. Why do the new docking affinities in Table 5 diff the old values in Table 3, assuming the same protein structures and docking protocol were used? In addition, I don’t know what the authors are trying to convey by saying that “the lowest affinity energy between carnosol and proteins as the best possibility of interaction (<7.0 Kcal/mol)” (page 12, lines 289-291).

First part: Why do the new docking affinities in Table 5 diff the old values in Table 3, assuming the same protein structures and docking protocol were used?

Answer

In both dockings, the oligomer of the proteins was considered for docking calculations, which led to different affinity values for PDB ID: 1DGB between Table 2 and Table 5 (initially proposed). Once the reviewers' responses were received, some modifications were made to the calculations shown in Table 5, which are described below.

  1. The affinity values shown in Table 5 initially (Exhaustiveness of 15) were recalculated with Exhaustiveness of 32 to improve the robustness of the calculations (See line 155).

  1. These new calculations were performed using only one structural unit, which is indicated at the end of the protein interaction residues. This chain was selected according to the data recorded in Protein Data Bank from the native ligand binding site, except for JUN (1A02), AHR (5NJ8), CASP3 (7RNF), and TNF (1TNR) which do not record information.

  1. Protein residues mediating the interaction with carnosol, identified with Protein-Ligand Interaction Profiler (PLIP) (https://plip-tool.biotec.tu-dresden.de/plip-web/plip/index (see line 157 and 159) are shown in the last column of Table 5.

Answer

These new calculations led to a slight variation of the results initially shown in Table 5 and Figure 7.

Second part: In addition, I don’t know what the authors are trying to convey by saying that “the lowest affinity energy between carnosol and proteins as the best possibility of interaction (<7.0 Kcal/mol)” (page 12, lines 289-291).

Answer

This was intended to convey that the lowest binding energy represented the best possibilities for interaction. Due to confusion, the sentence was replaced by “Considering the lowest affinity energy as the best interaction possibility, HSP90AA1, MAPK1 and MAPK3 proteins have the highest potential to be carnosol targets. Carnosol binding to HSP90 is supported by the hydrophobic interactions with LEU107A, PHE138A, VAL150A, TRP162A and the hydrogen bonds with ASN51A, TYR139A and PHE138A (Figure 7A). The binding of carnosol to MAPK1 is supported by the hydrophobic interactions with ILE31A, LYS54A, ILE84A, GLN105A, LYS114A, LEU156A, and the hydrogen bonds with LYS54A and ASP111A (Figure 7B). Carnosol was coupled to 6GES through of hydrophobic interactions with TYR53A, LYS71A, ILE73A, LEU187A, and the hydrogens bonds with TYR53A, and ASP184A (Figure 7C)” (See lines 317 to 326).

2) In Fig. 7, the authors display the binding modes of carnosol to HSP90, MAPK1, and MAPK3, whose affinities respectively rank 1, 3, and 4 in Table 5. Then what about the binding to CAT, which has the 2nd highest affinity?

Answer

Due to the fact that oligomers were initially considered in all calculations (Table 5), Carnosol bound to CAT residues in several chains, which prevented showing a single binding site since when forming the complex, the interactions did not correspond to a single chain, contrary to what happened for HSP90AA1, MAPK1 and MAPK3.

With the adjustments made to the calculations shown in Table 5 (see previous answer), where the completeness is improved and we work with a single monomer, the binding affinity between carnosol and CAT was the fourth best affinity, below carnosol and HSP90AA1, MAPK1 and MAPK3, which are shown in Fig. 7.

  1. Other comments:

1) Where are the supporting figures and tables? I don’t see them anywhere.

Answer

See Supplementary Material file.

2) Table 3: the 2nd column should be “PDB ID”, the 3rd column should be “Gene”, the “Uniprot ID” information is missing.

Answer

Table 3 was adjusted according to the observations. The first column as "No.", the second "PDB ID", the third "Gene", the fourth "Uniprot ID", and the last one, "description".

Round 2

Reviewer 1 Report

Dear Authors,

1. There is a pre-defined option in Pubchem and a format to cite the structures, please follow the guidelines for the same. Please do not mention the URL in the manuscript.

2. In results of MD simulation, "minimum and maximum RMSD obtained by MAPK1-6H3 were 0.23 and 4.65 nm, and by MAPK1-carnosol 0.31 and 6.91 nm, respectively"

I find the values maximum values very unrealistic and not coinciding with the RMSD plot provided. An RMSD change of 4.65nm or 46.5 Angstrom in not possbile in the current scenario.

3. The RMSD plot for MAPK1 with both the ligands doesnt seem to be equilibrated and stabalized. Please provide radius of gyration plot for better understanding of protein ligand interaction.

4. In the methods section of MD simulation, Please provide the time for recording interval.

5. Provide more information on the tool used to perform GBSA studies and cite it.

Author Response

There is a pre-defined option in Pubchem and a format to cite the structures, please follow the guidelines for the same. Please do not mention the URL in the manuscript.

Reference 22 corresponds to the 3D structure of Carnosol (see lines 125, 543 and 544).

In results of MD simulation, "minimum and maximum RMSD obtained by MAPK1-6H3 were 0.23 and 4.65 nm, and by MAPK1-carnosol 0.31 and 6.91 nm, respectively"

Reviewing the image, an error was noticed in the y-axis scale, which was made when vectorizing the graph. The vectorized image was adjusted to the real values. (See Fig. 8)

I find the values maximum values very unrealistic and not coinciding with the RMSD plot provided. An RMSD change of 4.65nm or 46.5 Angstrom in not possbile in the current scenario.

Reviewing the image, an error was noticed in the y-axis scale, which was made when vectorizing the graph. The vectorized image was adjusted to the real values. (See Fig. 8)

The RMSD plot for MAPK1 with both the ligands doesnt seem to be equilibrated and stabalized. Please provide radius of gyration plot for better understanding of protein ligand interaction.

Radius of gyration plot is showed in Fig. S2. and the description of the results are shown in lines 362 to 374

In the methods section of MD simulation, Please provide the time for recording interval.

The trajectories were saved every 0.01ns (See lines 197 and 198)

Provide more information on the tool used to perform GBSA studies and cite it.

More information on the tool used to perform GBSA studies is provided on lines 199 to 205.

Reviewer 2 Report

In the revised manuscript, the authors have appropriately addressed all my questions and concerns. I believe the manuscript is in good shape and publishable.

Author Response

In the revised manuscript, the authors have appropriately addressed all my questions and concerns. I believe the manuscript is in good shape and publishable.

Answer

Thanks a lot.  We appreciate your great comments.